# Structure–Function Diversity of Calcium-Binding Proteins (CaBPs): Key Roles in Cell Signalling and Disease

**DOI:** 10.3390/cells14030152

**Published:** 2025-01-21

**Authors:** Vanessa S. Morris, Ella M. B. Richards, Rachael Morris, Caroline Dart, Nordine Helassa

**Affiliations:** Department of Biochemistry, Cell and Systems Biology, Institute of Systems, Molecular and Integrative Biology, Faculty of Health and Life Sciences, University of Liverpool, Liverpool L69 3BX, UK; v.s.morris@liverpool.ac.uk (V.S.M.); e.richards@liverpool.ac.uk (E.M.B.R.); rachael.morris@liverpool.ac.uk (R.M.); cdart@liverpool.ac.uk (C.D.)

**Keywords:** calcium-binding proteins, calcium signalling, structural biology, interactions, ion channels, calcium channels

## Abstract

Calcium (Ca^2+^) signalling is a fundamental cellular process, essential for a wide range of physiological functions. It is regulated by various mechanisms, including a diverse family of Ca^2+^-binding proteins (CaBPs), which are structurally and functionally similar to calmodulin (CaM). The CaBP family consists of six members (CaBP1, CaBP2, CaBP4, CaBP5, CaBP7, and CaBP8), each exhibiting unique localisation, structural features, and functional roles. In this review, we provide a structure–function analysis of the CaBP family, highlighting the key similarities and differences both within the family and in comparison to CaM. It has been shown that CaBP1–5 share similar structural and interaction characteristics, while CaBP7 and CaBP8 form a distinct subfamily with unique properties. This review of current CaBP knowledge highlights the critical gaps in our understanding, as some CaBP members are less well characterised than others. We also examine pathogenic mutations within CaBPs and their functional impact, showing the need for further research to improve treatment options for associated disorders.

## 1. Calcium (Ca^2+^) as a Cellular Messenger: Signal Sources and Regulatory Mechanisms

Ca^2+^ ions serve as essential and highly versatile cellular messengers across all cell types, regulating diverse activities in response to both extrinsic and intrinsic stimuli [1]. The nature and dynamics of Ca^2+^ signals are finely tuned to each cell type’s specific physiological needs. Extrinsic stimuli that trigger Ca^2+^ signalling include, but are not limited to, hormonal, neurotransmitter, electrical, pH changes, cytotoxic agents, and temperature [2]. Intrinsic and extrinsic stimuli typically result in alterations of the intracellular Ca^2+^ concentrations from the normal cytosolic resting concentration (~100 nM) [3]. Intrinsic stimuli means that the signal is coming from within the cell itself rather than an outside factor, such as temperature, causing an alteration from the resting state [3]. One example of an intrinsic cue includes the spontaneous Ca^2+^ signals that are observed in cardiac myocytes, in which an increase in the subsarcolemmal intracellular Ca^2+^ concentration occurs simultaneously with the end of diastolic depolarisation due to the local release of Ca^2+^ from the sarcoplasmic reticulum (SR) [4]. Spontaneous Ca^2+^ signals have also been observed in developing neurons, though these were found to be also dependent on extracellular Ca^2+^ and neural network activity when investigated in mice [5]. The nature and source of these stimuli determine the magnitude of the change in cytosolic Ca^2+^ levels relative to the resting state. Importantly, a sustained increase in Ca^2+^ levels can cause acute organelle remodelling, the production of reactive oxygen species, mitochondrial permeability transitions, and the activation of Ca^2+^-dependent proteases [3]. As an example, significant increases in Ca^2+^ concentrations can trigger excitotoxicity of the neurons via the disruption of the Ca^2+^-mediated regulation of the glutamatergic synapses, leading to glutamate-induced cell death [6]. Additionally, alterations to the levels of Ca^2+^ in vascular smooth muscle cells, which are responsible for muscle contraction and serve as a negative feedback signal to induce vasodilation [7], can lead to cell death [8,9]. Therefore, the precise regulation of Ca^2+^ signalling is essential to facilitate the timely return to resting levels following any perturbation.

Cells have several organelles which contribute to the regulation of Ca^2+^ signalling, including the mitochondria [10], lysosomes [11], and endoplasmic reticulum (ER), although the two primary sources of Ca^2+^ in most cells are the release of Ca^2+^ from intracellular stores (e.g., the ER) and the influx of Ca^2+^ from the extracellular space [2]. Although both mechanisms can be utilised by cells, the levels that each cell uses fluctuate and vary between cell type and localisation. For example, cardiac muscle cells rapidly cease contracting when extracellular Ca^2+^ is removed, while skeletal muscle cells can continue to contract under similar conditions [12,13]. Cells mobilise Ca^2+^ from intracellular stores, such as the ER, through channels that span the membranes of organelles, such as inositol 1,4,5-triphosphate receptors (IP3Rs) and ryanodine receptors (RyRs) [14,15]. Extracellular Ca^2+^ enters cells via the opening of specific ion channels, such as the voltage-gated Ca^2+^ channels (VGCCs) [16] and ligand-gated ion channels (LGICs), which are activated by ligands, such as glutamate, gamma-aminobutyric acid (GABA), and acetylcholine [16]. The activity of the channels involved in both intracellular and extracellular Ca^2+^ influx are tightly regulated by a range of mechanisms. These mechanisms are regulated by specific proteins, such as calmodulin (CaM), neuronal Ca^2+^ sensor-1 (NCS-1), and Ca^2+^-binding proteins (CaBPs).

The focus of this review is on the Ca^2+^-binding protein (CaBP) family, for which little information is available. We provide a comprehensive overview of the CaBP family, highlighting their localisation, structure, interactions, and clinically relevant mutations. This may offer insights into the characteristics and mechanisms of the lesser-studied family members.

## 2. CaBPs: Localization and Expression

CaBPs form a subfamily of CaM-like proteins that play a critical role in regulating ion channels (e.g., Ca_v_1.2) and intracellular trafficking enzymes (e.g., phosphatidylinositol 4-kinase III β, PI4KIIIβ) [17]. The CaBP family has been shown to have co-evolved with vertebrate animals and share a similar domain organisation to CaM, with four EF-hand motifs [18]. EF-hand motifs are helix–loop–helix structural domains often found within Ca^2+^-binding proteins, where the N-terminal helix “E” is connected to the C-terminal helix “F” by a looped structure around the Ca^2+^ ion. CaBPs share ~46–58% genetic sequence similarity with CaM and can mimic its interactions with ion channels and other targets [19]. Unlike CaM, which is a ubiquitously expressed protein, CaBPs are enriched within neuronal tissue, where it is thought that they are crucial to Ca^2+^ homeostasis [20]. The CaBP family includes six members: CaBP1 (also known as caldendrin), CaBP2, CaBP4, CaBP5, CaBP7 (also known as calneuron II), and CaBP8 (also known as calneuron I). Notably, *CABP3* is considered a pseudogene with no detectable protein product, and no *CABP6* gene has been identified in humans [18]. Previous studies have shown that CaBPs can be divided into two subfamilies, according to their function and expression patterns: one group consists of CaBP1, 2, 4, and 5, while the other consists of CaBP7 and 8 [18]. Despite the different evolutionary pathways of the CaBPs from either CaBP1 or CaBP8, their sequence alignment reveals substantial homology among all CaBPs and with CaM [20].

While all CaBPs are essential for Ca^2+^ signalling in humans, they differ in their precise roles in this process. These differences are reflected in the subcellular localisation and expression patterns seen between the different proteins, as summarised in Table 1. Some CaBPs, such as CaBP5, CaBP7, and CaBP8, have been studied less extensively than CaBP1 or CaBP4 [19]. For example, there is limited evidence regarding the subcellular localisation of CaBP5 [21]. Although there are overlaps in localisation between all the CaBPs and CaM, some proteins exhibit more specific localisations related to their function. For example, CaBP7 and 8 are predominantly found at the trans-Golgi network membrane [22]. Furthermore, protein and mRNA expression levels vary by cell type, reflecting the specific functions of each individual CaBP. CaBP4, for instance, is highly expressed in the rod and cone photoreceptor cells within the retina, aligning with the expression pattern of L-type Ca^2+^ channel Ca_v_1.4, which CaBP4 regulates [21]. This channel plays a crucial role in the development and function of photoreceptor synapses, and its dysfunction has been linked to various visual acuity disorders [23]. Thus, the elevated expression of CaBP4 in photoreceptor cells is consistent with its role in the visual pathway [21]. In contrast, CaM is ubiquitously expressed across most cell types, ranging from neuronal to cardiomyocytes, at relatively uniform levels [24]. Overall, CaBPs are less uniformly abundant than CaM, indicating the greater specificity of their roles, particularly in neuronal cell types [24].

## 3. Structural Analysis of CaBPs

Despite differences in their expression patterns and localisations, the CaBP family is thought to show structural similarities to one another, but also to CaM in both Ca^2+^-free and Ca^2+^-bound forms. CaM’s structure has been previously described as a dumbbell-shaped molecule which contains two lobes connected by a very flexible central linker [38]. Each of the CaM lobes contains two Ca^2+^-binding EF-hand motifs, giving four binding sites in total [39]. When these EF-hand motifs in CaM bind with Ca^2+^ ions, it causes conformational changes in which the protein adopts an ‘open’ conformation [38]. Similar responses to Ca^2+^ have also been observed in troponin C and CaBPs, although the extent of the conformational changes and the number of ions bound to the protein can vary [40,41]. CaBPs also contain four EF-hand motifs that are arranged in a similar lobe structure connected by a linker as in CaM. However, many CaBPs contain an additional N-terminal domain [38]. Upon Ca^2+^ binding, CaBPs undergo significant structural changes, although they have alterations in their second EF-hand that prevent binding at this site [19].

### 3.1. Computational Structural Predictions

The AlphaFold protein structure database, developed by Google DeepMind, utilises an AI system to predict the 3D structure of proteins based on their amino acid sequence [42]. Currently, this tool contains predictions for over 214 million protein structures. The database allows for a comparison of CaBPs structural predictions, both among themselves and with other proteins with similar functions. CaBP1, CaBP2, CaBP4, and CaBP5 exhibit a high degree of similarity to each other and to CaM. Notably, three of these proteins (CaBP1, CaBP2, and CaBP4) contain an N-terminal region of varying length with low model confidence (pLDDT < 50), suggesting that this region is unstructured and/or disordered [43]. This observation is supported by the predicted aligned error (PAE) scores, which indicate a high expected error prediction for the amino acids in the N-terminal regions, reinforcing the notion of an unstructured region. In contrast, the predictions for CaBP5 show no unstructured N-terminal region. Despite these differences, the PAE diagrams for CaBP1–5 reveal two distinct regions with low predicted error, approximately equal in size. This suggests that after the disordered N-terminal region, CaBP1–5 are organised into two lobes with a strong similarity to the two-lobe structure observed in CaM [38].

The model’s confidence for the entire structures of CaBP7 and CaBP8 is significantly lower than that for CaBP1–5, indicating a lower level of similarity when compared to experimentally solved protein structures. This suggests that, unlike CaBP1–5, CaBP7 and CaBP8 do not share a substantial structural similarity with CaM. Previous studies have also supported the notion that CaBP7 and CaBP8 are structurally distinct from the other CaBPs, as they are believed to have evolved separately from this family [18]. This is possibly due to the different localisation and function of CaBP7 and 8, as they are involved in vesicle trafficking, compared to CaBP2, for example, which has been linked to the auditory pathway. Although CaBP7 and CaBP8 form a distinct subfamily of CaBPs [20], the AlphaFold predictions reveal some structural similarities with CaBP1–5. Notably, the unstructured N-terminal region predicted for CaBP1, CaBP2, and CaBP4 is also present in both CaBP7 and CaBP8. Additionally, following this unstructured region, CaBPs contain helical regions organised into lobes.

### 3.2. Experimentally Determined 3D Structures

The structures of the various CaBPs, as well as other key Ca^2+^-binding proteins, such as CaM and the neuronal Ca^2+^ sensors (NCSs), can be investigated using nuclear magnetic resonance (NMR) or X-ray crystallography [44]. These methods enable the determination of the protein structures in both Ca^2+^-free and Ca^2+^-bound states, thereby facilitating an understanding of the key conformational changes that occur upon Ca^2+^ binding. Databases, such as the Biological Magnetic Resonance Data Bank (BMRB) and Protein Data Bank (PDB), compile these NMR and crystallographic structures, providing valuable resources for researchers [45,46].

#### 3.2.1. Three Dimensional (3D) Structures in Ca^2+^-Free Conditions

To date, there are no Ca^2+^-free NMR structures for CaBP2, CaBP5, or CaBP8. A summary of the deposits for the CaBPs and CaM is presented in Table 2 and Figure 1. CaM is included due to the similarities in the computational modelling predictions and structure–function relationship between CaM and CaBPs [19,47,48]. Consistent with the AlphaFold predictions, NMR studies have shown that CaBP1 has a Ca^2+^-free structure similar to that of CaM [49]. CaBP1 and CaM are both key regulators of ion channels, such as voltage-gated calcium channels (VGCCs). Thus, the structural similarities observed between the two further support the idea of them having a similar structure–function relationship. Although the first 10 residues of CaBP1 could not be assigned, over 95% of the backbone and more than 75% of the side chains of human CaBP1 have been successfully assigned [50]. Notably, the CaBP1 structure in the database was obtained in the presence of Mg^2+^, as in other Ca^2+^-free experiments, which might have introduced structural variations [51].

The Ca^2+^-free structure of CaBP4 has also been studied using NMR; however, this was performed using murine CaBP4 rather than human [54]. Despite the use of a different species, the NMR-derived CaBP4 structure closely matched the predictions made by AlphaFold, including the observation of a disordered N-terminal region that was removed prior to the NMR analysis [54]. The experiments on Ca^2+^-free CaBP4 were performed in EDTA or MgCl_2_, reflecting its magnesium (Mg^2+^)-bound state in light-activated photoreceptor cells under reduced cytosolic Ca^2+^ levels [56]. However, it was observed that Mg^2+^-bound CaBP4 (100–271) was insufficiently stable for a structural analysis via NMR, therefore the N- and C-lobes were examined separately [54]. The interhelical angles of Mg^2+^-bound CaBP4 were similar to those of Ca^2+^-free CaM in some EF-hands, except EF-hand 2, which exhibited a notable variation (9.9 degrees larger in CaBP4) [52,54]. Therefore, once the unordered N-terminal region was removed, *Mus musculus* CaBP4 and *Xenopus laevis* CaM showed strong structural similarities. Although these are not human proteins, the high conservation of CaBPs within vertebrates suggests minimal structural variations between species [20]. Based on the computational predictions and evolutionary diversity, CaBP2 and CaBP5 are likely to share structural similarities with CaBP1 and CaBP4 [17,18,20]. This is further supported by the fact that CaBP1–5, despite variations in their tissue expression patterns, have all been seen to play a crucial role in the regulation of ion channel activity. The expected common structural features among CaBP1–5 would therefore be an organisation into two lobes connected with a linker and a disordered N-terminal region (except for CaBP5).

Unlike CaBP1–5, CaBP7 shows significant differences in both structure and function when compared to CaM. CaBP7 shares some structural similarities with the CaBP1–5 subfamily, particularly regarding the influence of Ca^2+^ binding on its conformation. Notably, CaBP7 retains a folded structure even in the absence of Ca^2+^ [55]. However, the NMR structure of CaBP7 does not cover the full sequence, limiting both the structural understanding of CaBP7 and detailed comparisons with other CaBPs [55]. Although there is currently no available NMR structure for CaBP8 in Ca^2+^-free conditions, it is likely to resemble the structure of CaBP7 because of their shared localisation, functions, and evolutionary pathway [18,20,22].

#### 3.2.2. Three Dimensional (3D) Structures in Ca^2+^-Bound Conditions

The NMR structure database enables access to Ca^2+^-bound structures for CaBP1, CaBP4, and CaBP7. The other CaBPs lack Ca^2+^-bound NMR structures within the database. A summary of the Ca^2+^-bound NMR and crystal structures is presented in Table 3 and Figure 2. The Ca^2+^-bound structure of CaBP1, similarly to its Ca^2+^-free form, has high assignment coverage, with over 95% of the backbone and most of the side chains identified [40,50]. The first 15 residues, which exhibit a disordered conformation, are not assigned. In the Ca^2+^-bound state, the CaBP1 NMR structure reveals three Ca^2+^-binding sites at EF-hands 1, 3, and 4 [40]. Notably, EF-hand 1 retains a closed conformation when bound to Ca^2+^, resembling the regulatory N-domain of troponin C [40,41]. The regulatory N-domain of troponin C moves dynamically during activation, playing a key role in cardiac muscle regulation. However, it remains unclear whether this mechanism is mirrored in CaBP1’s function. The closed conformation of the N-lobe is observed in both the NMR and crystal structures, with no Ca^2+^ bound at EF-hand 1 or 2 [49]. In contrast, EF-hands 3 and 4 show a more “open” conformation, similar to what has been observed in Ca^2+^-bound CaM [49,57,58]. This suggests that the inactive state of the second EF-hand in the N-lobe of CaBP1 prevents Ca^2+^ ions from binding, thereby restricting conformational change. Other CaBPs sharing this feature with CaBP1 may show similar structural variations in the presence of Ca^2+^.

The Ca^2+^-bound CaBP4 structure was determined using *Mus musculus* rather than *Homo sapiens* CaBP4, with the unordered region removed to facilitate the structure solution [54]. Under these conditions, the CaBP4 structure closely resembles that of CaBP1 and CaM, showing a conformational change upon Ca^2+^ binding [58]. Similar to CaBP1, CaBP4 binds to three Ca^2+^ ions at EF-hands 1, 3, and 4 [40]. While Ca^2+^ binding at EF-hand 1 is not sufficient to induce an open conformation for the N-lobe, the binding at EF-hands 3 and 4 results in an open conformation in the C-lobe, as observed for CaBP1 [40,54]. The interhelical angles in CaBP4 EF-hands 3 and 4 are notably smaller (104.1 and 88.3 degrees) than those in EF-hands 1 and 2 (134.7 and 141.1 degrees) [54]. Furthermore, the interhelical angles for EF-hands 1 and 2 in Ca^2+^-bound CaBP4 are similar to those in the Mg^2+^-bound state (133.8 and 140.7 degrees) [54], reinforcing that these EF-hands do not undergo conformational changes in the presence of Ca^2+^.

There are currently no NMR or crystal structures for *Homo sapiens* CaBP2 or CaBP5. However, an X-ray structure is available for *Entamoeba histolytica* CaBP5 (EhCaBP5), which shows two clear globular lobes connected by loops. EhCaBP5 has been linked to the initiation of phagocytosis of human erythrocytes. The structure of CaBP5 reveals a single Ca^2+^ ion bound at the N-terminal lobe [60], suggesting that the second EF-hand remains inactive [40,54]. Additionally, it is likely that for both CaBP2 and CaBP5, some or all of the Ca^2+^-binding EF-hands undergo significant conformational changes, adopting the “open” form of the C-lobe, as observed in other CaBPs [58].

In line with the computational predictions and NMR data for the Ca^2+^-free state, CaBP7 shows structural differences when compared to other CaBPs and CaM. CaBP7 undergoes significant structural changes upon Ca^2+^ binding, adopting an open conformation in the N-lobe, similar to that observed in the C-lobe of CaBP1, CaBP4, and CaM [55,58]. However, it features a larger exposed hydrophobic surface, suggesting notable structural distinctions from CaBP1–5 [55]. The available NMR structure for CaBP7, despite being from *Homo sapiens*, is incomplete, as only part of the sequence is included. This partial structure may lack key elements necessary for understanding CaBP7’s characteristics in comparison to other CaBPs. Additionally, the truncation may disrupt the structure if intra-molecular bonds are interrupted, a known issue when examining truncated isoforms [61]. Currently, no Ca^2+^-bound NMR or crystal structure is available for CaBP8. However, given its shared localisation, function, and evolutionary pathway with CaBP7 [18,20,22], CaBP8 is likely to exhibit a similar Ca^2+^-dependent structural response and a distinct conformation compared to CaBP1–5 and CaM.

## 4. Regulation of Ion Channel Activity by CaBP1–5

The CaM superfamily of Ca^2+^-binding proteins is defined by the highly conserved EF-hand motifs [62]. Closely related proteins such as CaBPs share domain similarities and organisation of the EF-hands with CaM [19,63], but differ in their sequence and functionality [64]. The shared targets of CaM and CaBPs are predominantly ion channels, specifically Ca^2+^ permeable. Despite their structural and evolutionary similarities [18], CaM and CaBPs can have differential regulatory effects on ion channels. In this review, we focus on the regulation by CaBPs of VGCCs, TRPC5, IP3Rs, and PIEZO2. Ryanodine receptors (RyRs) were initially considered for inclusion, as these are a widely expressed family of intracellular Ca^2+^ channels and have been shown to interact with other Ca^2+^-signalling proteins, such as CaM [65,66,67,68,69,70,71]. However, it has been shown that CaBP1 has no impact on RyRs activity [28] and there is no strong evidence to support a direct interaction between CaBPs and RyRs.

### 4.1. Voltage-Gated Ca^2+^ Channels (VGCCs)

VGCCs are membrane-localised channels that open in response to cell membrane depolarisation and are mediated by Ca^2+^-binding proteins. There are three main types of VGCCs expressed in the brain and heart, known as the Ca_v_1, Ca_v_2, and Ca_v_3 families [1,72]. VGCCs, also known as Ca_v_ channels, are heteromultimeric protein complexes composed of accessory β and δ subunits that flank a pore-forming α subunit. The α subunit contains four transmembrane regions and is bracketed by cytoplasmic N- and C-terminal domains [73]. The N-terminus contains a CaM-binding domain called the N-terminal spatial Ca^2+^ transforming element (NSCaTE). The NSCaTE is believed to be crucial for CaM to stabilise the closed state of the channel at higher concentrations of Ca^2+^ [74,75]. The C-terminal (known as CT1) also plays a crucial role in the regulation of these channels and contains important sites for interaction with both CaM and CaBPs [76,77]. One of these key sites is the IQ domain, known to bind to both CaM and CaBP1 [78,79,80]. The diverse expression patterns, cellular localisation, and presence of splice isoforms enable CaBPs to modulate Ca^2+^ channel activity in a highly specialised manner within the central nervous system. This modulation occurs both in conjunction with and opposition to CaM. Importantly, there is no functional redundancy with CaM despite their structural similarity. Instead, the regulatory roles of CaBPs remain distinct and finely tuned to their unique properties [49,81,82,83,84].

Ca_v_ channels rely on proteins like CaM [79,85] and CaBP1 [80,86] to finely tune their activity in response to subtle changes in Ca^2+^ concentration. The effects of these interactions vary significantly among the channel subtypes. The most well-known interactions of the CaBP family are with L-type VGCCs (Ca_v_1.1, 1.2, 1.3, and 1.4). CaBP1’s interaction with the L-type (Ca_v_1) IQ motif was previously characterised and revealed a strong structural similarity with the CaM-IQ interaction [87] (Figure 3). A structural model was proposed based on homology modelling using the crystal structure of the CaM-IQ complex [88]. This model suggests that the Ca^2+^-bound C-lobe of CaBP1 interacts with residues I1654 and Y1657 within the IQ motif [89]. However, further structural studies are necessary to validate this model. The dissociation constant (*K*_d_) for the interaction between Ca^2+^-bound CaBP1 and the Ca_v_1.2 IQ motif is reported to be 2.9 × 10^−10^ ± 7.0 × 10^−11^ M [87].

CaBP1 binding to Ca_v_1.2 has been shown to abolish CDI (Ca^2+^-dependent inactivation) and increase channel open probability, which can lead to CDF (Ca^2+^-dependent facilitation) [49]. This is consistent with previous studies showing that increased CaBP1 expression abolishes the CDI of Ca_v_ in certain neuronal cell types [80,87]. Within a cell, the apo-states of CaBP1 and CaM compete for binding to the IQ domain of Ca_v_1.2. This competition is thought to allow for the regulation of channel activity in response to Ca^2+^ signals, either resulting in CDI or CDF [87,90]. When CaM is bound to the IQ domain and cytosolic Ca^2+^ reaches higher levels, a conformational change occurs within CaM. The N-lobe of Ca^2+^/CaM binds to the NSCaTE domain of Ca_v_1.2 and promotes CDI [74,75] (Figure 3a). Alternatively, when CaBP1 is pre-associated with the IQ domain, increasing Ca^2+^ levels allow CaBP1 to remain bound to the IQ domain and to stabilise the open state of the channel (Figure 3b). Although multiple proposed Ca_v_ channel binding sites for CaBP1 have been identified [91], only CaBP1 binding to the IQ domain is thought to trigger CDF [90]. CaBPs interact with Ca_v_1.3 channels located in the inner hair cell (IHC) ribbon synapses [31]. Although CaBP1, 2, 4 and 5 are all expressed in the cochlea, CaBP1 was found to be the most efficient at blocking CDI [92]. The expression levels of CaBP4 and 5 are relatively low compared to CaBP1 and 2 [31,93]. The overexpression of CaBP2 strongly diminishes CDI in Ca_v_1.3 [31], aligning with recent findings from a CaBP1/2 knockout mouse model, which shows significantly enhanced CDI in Ca_v_1.3 [94]. Both CaBP1 and 4 interact with the CaM-binding sequence of Ca_v_1.3, but the effects of CaBP4 on CDI are weaker [92]. CaBP4 and 5 are known to interact with Ca_v_ channels in the retina, specifically at photoreceptor synapses and in rod and cone bipolar cells [33,95]. CaBP5 binds to Ca_v_1.2 in a Ca^2+^-dependent manner in the CT1 region, likely overlapping with CaM’s binding site due to the structural similarities between CaM and CaBPs [34]. CaBP4 is one of the most studied CaBPs and has been shown to interact with Ca_v_1.4. CaBP4 binding to Ca_v_1.4 is essential for neurotransmitter release in retinal photoreceptor cells [95,96]. Like other CaBPs, CaBP4 binds to the IQ motif (residues 1579–1605) of Ca_v_1.4, with a *K*_d_ of 0.8 ± 0.2 µM [54].

CaBP1 and CaM are the only proteins discussed in this review that also bind to other Ca_v_ channels, such as P/Q type channels (Ca_v_2.1). P/Q-type Ca^2+^ currents facilitate neurotransmitter exocytosis in the presynaptic zones of central neurons [97,98]. Unlike their effect on Ca_v_1 channels, where CaBP1 binding abolishes or reduces CDI, CaBP1 binding to Ca_v_2.1 only promotes CDI [82,83]. CDI is a critical negative feedback mechanism for Ca_v_ channels that limits Ca^2+^ entry following membrane depolarisation. In contrast to CaM, the binding of CaBP1 to Ca_v_2.1 causes a depolarising shift in the voltage dependence of activation, therefore reducing CDF and supporting faster channel inactivation [82]. This is achieved by CaBP1 binding to the CaM-binding domain of Ca_v_2.1 in a Ca^2+^-independent manner. This binding is believed to inhibit Ca^2+^ influx into the presynaptic terminal, thereby slowing neurotransmitter release and contributing to the regulation of synaptic plasticity [82].

### 4.2. Transient Receptor Potential Canonical Type 5 Channel

Transient receptor potential canonical type 5 (TRPC5) belongs to a subfamily of non-selective Ca^2+^-permeable channels that are highly expressed across multiple regions of the brain [99]. These channels are associated with a range of neuronal functions, including neuronal firing, synaptic transmission, neurite elongation, and growth cone guidance. TRPC5 is part of a group of TRPC channels with a similar sequence homology (TRPC1, 4, and 5), and forms tetrameric structures in cell membranes [100,101]. The regulation of TRP channels is predominantly achieved by Ca^2+^ binding and interaction with CaM (Figure 4) [102,103,104,105]. Notably, among the TRPC channels, only TRPC5 is known to interact with CaBPs (CaBP1), hence why only TRPC5 is focused on here. TRPC5 channels are specifically expressed in hippocampal neurons [106,107] and are activated by receptors coupled to phospholipase C (PLC) [108,109]. A disruption in TRPC5 transport or function negatively affects synaptogenesis and axon growth and the sites of action are likely to be associated with growth cone morphology and neurite extension [110]. Like TRPC5, CaBP1 is also abundantly expressed in hippocampal neurons [19]. The overexpression of CaBP1 in *Xenopus* oocytes was found to inhibit the TRPC5 channel in a dose-dependent manner, whereas the overexpression of CaM produced no significant effect on channel activity. However, in alternate studies, it has been demonstrated that CaM is essential for the Ca^2+^/CaM-mediated facilitation of the receptor-induced activation of TRPC5 channels [105]. In addition, an engineered triple mutant of CaBP1, which was unable to bind Ca^2+^, did not inhibit TRPC5 [84,111]. This suggests that CaBP1’s inhibitory effect on TRPC5 is Ca^2+^-dependent. Both the N- and C-termini of TRPC5 have shown binding with CaBP1. The C-terminus contains CaM-binding sites and has shown stronger binding than the N-terminus [84]. To narrow down the CaBP1-binding site on TRPC5, three residue sequences were deleted, reducing CaBP1 binding. However, the testing of short fragments containing only these sequences showed no binding, suggesting that CaBP1 binding to TRPC5 likely depends on conformational changes in the tertiary structure rather than specific linear sequences alone [84]. This interaction between CaBP1 and TRPC5 may have important functional consequences, as the neuronal presence of CaBP1 may reduce aberrant channel activity during transport from the somata to growth cone.

### 4.3. PIEZO2 Channels

PIEZO2 channels are mechanically activated and non-selective cation channels permeable to Ca^2+^, playing a key role in mediating ionic currents in primary sensory neurons [112,113,114]. They are large membrane proteins with over 2800 residues organised into 38 transmembrane helices with a fully closed pore flanked by both transmembrane and cytosolic constriction sites, contributing to its mechanical sensitivity and gating properties [114]. Although other CaBPs, such as 2, 4, and 5, have their expression limited to the brain, cochlea, and retina, CaBP1 is also expressed in peripheral neurons. Interestingly, a recent study demonstrated that CaBP1 inhibits PIEZO2 channels in dorsal root ganglion neurons (DRGNs) [115]. CaBP1’s inhibitory effect was particularly observed in large myelinated DRGNs, which play a critical role in transmitting information related to innocuous touch and are known to express high levels of CaBP1 [116,117]. Touch reception is mediated by mechanically activated (MA) receptors in DRGNs [118,119,120]. Based on CaBP1 expression patterns, it was hypothesised that CaBP1 could regulate MA channels across different classes of DRGNs. It was shown that CaBP1 negatively regulates PIEZO2, indicating its importance in controlling touch sensation. Mice lacking CaBP1 experienced tactile hypersensitivity with increased MA current densities in their peripheral neurons [115]. Moreover, co-immunoprecipitation detected via Western blotting suggested a direct protein–protein interaction between CaBP1 and PIEZO2. This provided the first evidence for Ca^2+^-sensor proteins such as CaBP1 as modulators of MA channels [115].

### 4.4. Inositol 1,4,5-Triphosphate Receptors

Ca^2+^ entry into cells is not exclusively from extracellular sources. Its release from internal stores, primarily the endoplasmic reticulum (ER), can regulate intracellular Ca^2+^ concentration ([Ca^2+^]i) [121]. Inositol 1,4,5-triphosphate receptors (IP3Rs), a family of channels localized to the ER membrane, mediate Ca^2+^ release into the cytoplasm. In the brain, Ca^2+^-binding proteins like CaM and CaBP1 regulate the open probability of IP3Rs, similarly to VGCCs [81,122]. Two distinct binding sites for CaM have been identified on IP3Rs: a low-affinity site on the N-terminus that binds CaM independently of Ca^2+^ [123,124], and a high-affinity site predominantly bound by Ca^2+^/CaM (Figure 5) [125]. CaBP1 is a high-affinity agonist for mammalian IP3R isoforms, binding with an approximately 100-fold greater affinity than CaM [29,81]. CaBP1 binds via its C-lobe to the cytosolic N-terminal of IP3R, within the IP3-Binding Core (IBC, residues 224–604) on the cytosolic N-terminal domain of IP3R. Upon Ca^2+^ binding, CaBP1 exposes hydrophobic residues (V101, L104, and V162) that engage with complementary hydrophobic residues (L302, I364, and L393) on the β-domain of the IBC (IBC-β), restricting intrasubunit movement between the IBC-β domain and the neighbouring suppressor domain (SD) of the IBC [126]. In this state, Ca^2+^-bound CaBP1 has an affinity (*K*_d_) of 3 ± 0.3 µM for the cytosolic N-terminal region of IP3R, when the receptor is saturated with IP3 [81]. IP3Rs naturally assemble into tetramers centred around an ion-conducting pore. Structural models of the IP3R N-terminus suggest that four CaBP1 molecules bind around the core via the hydrophobic residues on their C-lobes, forming a stabilizing ring around the binding core, similarly to how FKBP12 stabilises RyRs [127]. This interaction favours the closed state of the channel, thereby reducing IP3R activity [126]. IP3R regulation is crucial for Ca^2+^-induced Ca^2+^ release (CICR), a physiological phenomenon responsible for complex regenerative Ca^2+^ within cells. CaBP1 has demonstrated an effect on the open probability of IP3Rs as a negative feedback mechanism, implicating it as a key component of a Ca^2+^ regulatory pathway. To date, there is no direct evidence that other CaBPs bind to IP3Rs. CaBP2, which shares a 59.8% sequence homology with CaBP1 and is also neuronally expressed, shows functional overlap with CaBP1 in inner hair cells (IHCs) when interacting with Ca_v_1.3 channels [31], suggesting similar regulatory mechanisms for the two CaBPs. However, NMR experiments on Ca^2+^-bound CaBP4 have revealed a distinct structural environment for the key residues involved in CaBP1’s interaction with IP3R, providing insight into why CaBP4 does not bind IP3Rs [128].

A comparison of the known interactions between CaM, CaBP1–5, and ion channels are summarised in Table 4.

## 5. CaBP7 and CaBP8 (Calneurons): Localisation and Role in PI4KIIIβ Regulation

CaBP7 and CaBP8 form a distinct family of CaBPs due to their unique structural features [17,63,64]. Their C-terminal extensions enable them to specifically localise to the Golgi membrane, making them crucial for Golgi trafficking [17,130]. Unlike CaBP1–5, CaBP7 and CaBP8 lack an N-myristoylation motif for membrane anchoring. Instead, their strong association with the Golgi is facilitated by post-translational transmembrane insertion via their unique C-terminus [17,36,130]. This structural feature differentiates them from other CaBPs, and they are not known to interact with Ca^2+^ channels or the other ion channels discussed in this review.

To date, only two direct interactions have been identified for CaBP7 and CaBP8. They interact with the transmembrane domain recognition complex 40 (TRC40) [130] and with PI4KIIIβ, which has been shown to reduce phosphatidylinositol 4-phosphate (PI4P) production. PI4KIIIβ is a lipid kinase essential for Golgi-to-membrane trafficking and vesicle endo/exocytosis [131,132,133,134]. The Golgi apparatus, which can sequester and release Ca^2+^, also facilitates Ca^2+^ signalling, with signals being transduced directly or indirectly by CaBPs [135].

CaBP7 interacts with PI4KIIIβ via its N-terminal domain in a Ca^2+^-dependent manner. Upon Ca^2+^ binding, CaBP7 undergoes a conformational change that makes its N-terminus resemble the C-termini of CaBP1 and CaM, although distinct surface properties may allow CaBP7 to target specific proteins [55]. The N-termini of CaBP7 and CaBP8 share an 81% sequence identity, which potentially explains why CaBP8 can also regulate PI4KIIIβ [55]. NCS-1, another Ca^2+^-binding protein, promotes PI4KIIIβ activity and competes with CaBP7 and CaBP8 for binding. CaBP7 and CaBP8 binding to PI4KIIIβ has been shown to reduce PI4P production [55,136,137,138,139]. NCS-1 and CaBP7 and 8 act as a molecular switch, with calneurons suppressing PI4KIIIβ activity during resting or sub-optimal Golgi Ca^2+^-transients providing tonic inhibition. This regulates vesicle exocytosis and trafficking from the Golgi in neurons, potentially in postnatal development [136].

## 6. CaBPs Dysfunction Is Associated with Neuronal Disorders

Ca^2+^-sensing proteins such as CaBPs respond to complex and highly dynamic intracellular Ca^2+^ changes, which is crucial for controlling many cellular processes involved in neuronal homeostasis. Aberrant Ca^2+^ signalling due to dysfunctional CaBPs has been associated with various pathologies, mainly affecting neuronal and auditory pathways (Table 5, Figure 6, Figure 7 and Figure 8). The differential localisation and expression patterns amongst the CaBPs result in distinct cellular functions and, depending on which protein is affected, can give rise to different disease phenotypes. For example, CaBP1, a key player in synaptodendritic Ca^2+^ signalling, has shown abnormal expression levels in the post mortem brains of patients with chronic schizophrenia, which likely disrupted synaptodendritic signalling [140]. Additionally, the translocation of CaBP1 to the post-synaptic density has been observed in epileptic rat models and has been associated with the pathophysiology of seizure activity [141]. The CaBP1 R308X nonsense mutation, which results in a truncated version of the protein, was identified in a male of unknown age and has been linked to microcephaly [142] (Figure 6a). The CaBP7 T156M missense mutation has been associated with colon cancer [143] (Figure 8a). These observations highlight that there is no link between CaBPs and the clinical severity of a disease in terms of the phenotypic presentation. Primarily localised in inner hair cells, CaBP2 regulates auditory transmission through the inhibition of Ca_v_1.3 CDI [31]. Furthermore, many CaBP2 mutations are linked to autosomal recessive hearing loss [144,145,146,147,148,149] (Figure 6b). The CaBP2 E156X nonsense mutation, which results in a truncated version of the protein, completely abolishes Ca^2+^ binding to the C-lobe whilst having a 1.3-fold decrease in affinity for the N-lobe. E156X is also unable to effectively inhibit the CDI of Ca_v_1.3 due to losing its Ca^2+^-binding capabilities [146]. Another CaBP2 variant, F164X, causes a reading-frame shift and prematurely truncates the protein, such that it lacks the C-lobe’s EF-hands [148,149]. Like E156X, F164X displays weaker binding to Ca^2+^ and reduced inhibition of Ca_v_1.3 CDI, thereby suggesting that the truncation might interfere with its role as a Ca^2+^ sensor [148,149]. The majority of CaBP-associated diseases are related to auditory and neuronal dysfunction. CaBP2 mutations, which are commonly associated with inherited auditory impairment, have been diagnosed within a small subset of individuals via audiological testing (pure-tone audiometry, otoacoustic emissions, and auditory brainstem response) [148] and genetic sequencing techniques (whole-genome sequencing and whole-exome sequencing) [144,145,146,147,148,149]. Due to the complex relationships of genotypes to phenotypes and limited clinical/molecular studies, very little is known about CaBP2-associated congenital hearing loss in relation to prognosis and therapy. However, one study found that the c.637+1G > T *CABP2* mutation caused moderate-to-severe hearing loss within three Iranian families [148]. As there are no targeted CaBP2-associated congenital hearing loss treatments, individuals will likely require hearing aids/cochlear implants combined with speech and language therapy.

Mutations within CaBP4 are associated with a range of neuronal pathologies, such as congenital stationary night blindness [150,151,152], cone–rod dystrophy [151,152,153,154,155,156,157,158,159,160,161,162], and autosomal dominant nocturnal frontal lobe epilepsy (ADNFLE) [163,164] (Figure 7a). The literature identifying CaBP4 mutations are from whole-exome sequencing/genetic testing studies; thus, there is limited information at the molecular level about the structural–functional consequences of these mutations. Ocular disorders, such as congenital cone–rode dystrophy (progressive degeneration of the cone and rod photoreceptor cells in the retina) and congenital stationary night blindness (a rare, non-progressive disorder that primarily affects vision in low light or darkness), are associated with CaBP4 mutations. Visual testing, including an evaluation of the best-corrected visual acuity, kinetic Goldmann perimetry, slit-lamp examination, fundoscopy, and optical coherence tomography, along with genomic screening, aids in the diagnosis of CaBP4-associated inherited ocular disorders [151]. However, the location and type of mutation can give rise to different ocular phenotypes [150,151,152,153,154,155,156,157,158,159,160,161,162], which often proves challenging for a speedy diagnosis. Congenital cone–rode dystrophy is progressive and many of individuals’ vision will worsen over time; however, the severity and rate of progression vary depending on the genetic mutation [151,152,153,154,155,156,157,158,159,160,161,162]. Currently, there are no targeted therapies for congenital cone–rode dystrophy or congenital stationary night blindness, and they are generally managed via vision aids, lifestyle adaptations, corrective lenses, and low-light adaptations, respectively. There are no known clinically significant mutations within CaBP5 and CaBP8 that are associated with major diseases (Figure 7b and Figure 8b).

**Table 5 cells-14-00152-t005:** Summary of key clinical pathogenic mutations within CaBPs.

Gene	cDNA Mutation	Protein Change	Mutation Type	Phenotype	References
*CABP1*	c.922C > T	R308X	Nonsense	Microcephaly	[142]
*CABP2*	c.250G > A	E84K	Missense	Autosomal recessive hearing loss	[144]
	c.311G > A	-	Missense	Autosomal recessive hearing loss	[145]
	c.466G > T	E156X	Nonsense	Autosomal recessive hearing loss	[146]
	c.490-1G > T	-	Splice	Autosomal recessive hearing loss	[147]
	c.637+1G > T	F164X	Splice	Autosomal recessive hearing loss	[148,149]
*CABP4*	c.65del	P22fs	Frameshift	Congenital cone–rod dystrophy	[153]
	c.81_82insA	P28fs	Frameshift	Congenital cone–rod dystrophy	[152,154,159]
	c.145C > T	R49X	Nonsense	Congenial stationary synaptic dysfunction	[153,155]
	c.154C > T	R52X	Nonsense	Congenital cone–rod dystrophy	[152]
	c.366+1G > T	-	Splice	Congenital stationary night blindness	[150]
	c.370C > T	R124C	Missense	Congenital stationary night blindness	[151,152]
	c.464G > A	G155D	Missense	Nocturnal frontal lobe epilepsy	[163,164]
	c.646C > T	R216X	Nonsense	Congenital cone–rod dystrophy	[152,156,157,160,161]
	c.673C > T	R225X	Nonsense	Congenital cone–rod dystrophy	[158,161,162]
	c.757C > T	R253X	Nonsense	Congenital cone–rod dystrophy	[153]
	c.773A > T	N258I	Missense	Congenital cone–rod dystrophy	[157]
	c.800_801delAG	E267	Deletion	Congenital cone–rod dystrophy	[151,152,154]
*CABP7*	-	T156M	-	Colon cancer	[143]

## 7. Conclusions

CaBPs are a family of CaM-like regulators that play a crucial role in fine-tuning Ca^2+^ signalling. Unlike the ubiquitously expressed CaM, CaBPs show more specific expression patterns and functions within signalling pathways. Here, we summarised the available structural data on CaBPs and showed that CaBP1–5 share significant structural similarities in both Ca^2+^-bound and Ca^2+^-free states with each other as well as with CaM and troponin C. However, to confirm the structural similarities among the CaBPs, further experimental studies are needed, such as obtaining additional structural data on CaBP2, CaBP4, and CaBP5. In contrast, CaBP7 and CaBP8 display greater structural divergence from the other CaBPs and CaM, forming a distinct subfamily. This divergence is consistent with their unique expression patterns, interaction profiles, and specialised roles, which differ significantly from those of CaBP1–5. Despite their differences from other CaBPs, the evolutionary and structural analyses indicate that CaBP7 and CaBP8 are closely related to each other. However, further experimental structural studies are needed to better understand this subfamily. Currently, only partial structural data are available for CaBP7, and no structural data exist for CaBP8.

We have highlighted the crucial roles of CaBP1–5 in regulating ion channels, focusing on Ca_v_ channels, IP3R, TRPC5, and PIEZO2. Notably, CaBP1, similarly to CaM, has been shown to interact with all these channels except Ca_v_1.4. In contrast, the other CaBPs demonstrate greater specificity, interacting with channels that align closely with their localisation and functional roles in Ca^2+^ signalling. For example, CaBP4 primarily interacts with Ca_v_1.3 and Ca_v_1.4, both of which are highly expressed in the retina. Additional work is needed to investigate other potential ion channel interactions for CaBP1–5. This is particularly relevant for CaBP2 and CaBP5, which have been investigated less extensively than CaBP1 and CaBP4. Considering the similarities between the interaction partners of CaBP1 and CaM, it may be worth investigating other ion channels regulated by CaM for any overlap. Currently, there is no evidence supporting CaBP7 or 8 regulating or interacting with ion channels. To date, CaBP7 and CaBP8 have only been shown to interact with PI4KIIIβ and TRC40, which is essential for their role in vesicle trafficking and maintaining secretory pathway function.

Clinically relevant mutations in CaBPs are associated with a range of pathogenic phenotypes, including autosomal recessive hearing loss caused by mutations in CaBP2, as well as various visual disorders, dystrophies, and frontal lobe epilepsy linked to CaBP4 mutations. However, the underlying mechanisms of many of these mutations remain poorly understood, often resulting in ineffective treatment options. To improve outcomes, further research is needed to elucidate the precise molecular and cellular impacts of these mutations and their contributions to the observed pathogenic phenotypes. Additionally, the search for new pathogenic mutations in the CaBP family should continue, either by assessing known mutations for their pathogenic potential or identifying novel mutations using advanced screening methods. Future research should aim to fill the current knowledge gaps in the structure–function relationship of CaBPs, as this would lead to a better pathological understanding and enable the development of more effective, targeted therapeutic strategies.

## Figures and Tables

**Figure 1 cells-14-00152-f001:**
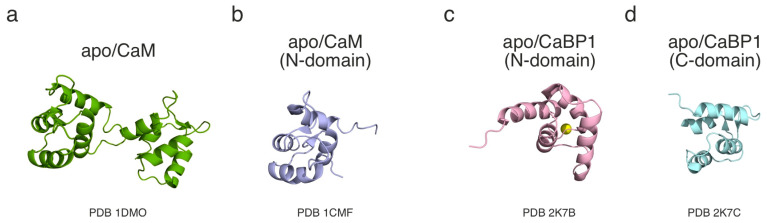
Cartoon representation of the 3D structures of apo/CaM and apo/CaBP1. (**a**) Full-length apo/CaM (PDB 1DMO), (**b**) N-domain of apo/CaM (PDB 1CMF), (**c**) N-domain of apo/CaBP1 (PDB 2K7B), and (**d**) C-domain of apo/CaBP1 (PDB 2K7C). apo/CaBP1 does not structurally align with apo/CaM. Mg^2+^ ion is represented by yellow sphere.

**Figure 2 cells-14-00152-f002:**
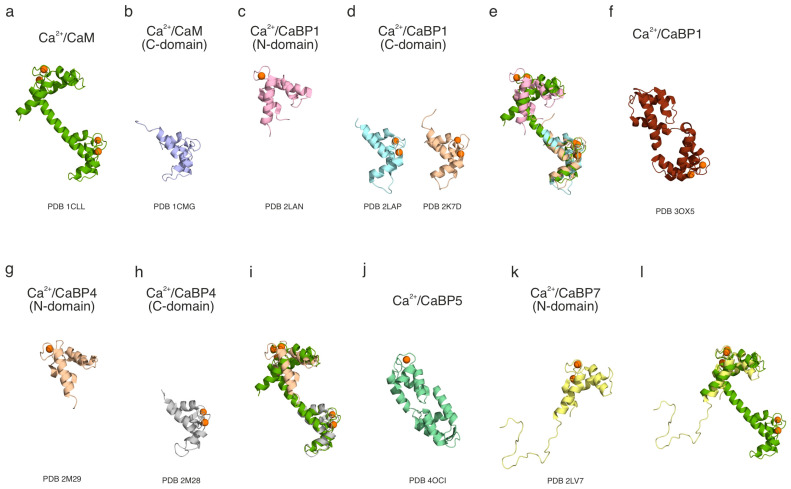
Cartoon representation of the 3D structures of Ca^2+^/CaBPs and their comparison with Ca^2+^/CaM. (a) Full-length Ca^2+^/CaM (PDB 1CLL), (**b**) C-domain of Ca^2+^/CaM (PDB 1CMF), (**c**) N-domain of Ca^2+^/CaBP1 (PDB 2LAN), (**d**) C-domain of Ca^2+^/CaBP1 (PDB 2LAP, PDB 2K7D), (**e**) structural alignment of Ca^2+^/CaM and Ca^2+^/CaBP1 structures, (**f**) full-length Ca^2+^/CaBP1 (PDB 3OX5), (**g**) N-domain of Ca^2+^/CaBP4 (PDB 2M29), (**h**) C-domain of Ca^2+^/CaBP4 (PDB 2M28), (**i**) structural alignment of Ca^2+^/CaM and Ca^2+^/CaBP4 structures, (**j**) full-length Ca^2+^/CaBP5 (PDB 4OCI), (**k**) N-domain of Ca^2+^/CaBP7 (PDB 2LV7), and (**l**) structural alignment of Ca^2+^/CaM and Ca^2+^/CaBP7 structures. Ca^2+^ ions are represented by orange sphere.

**Figure 3 cells-14-00152-f003:**
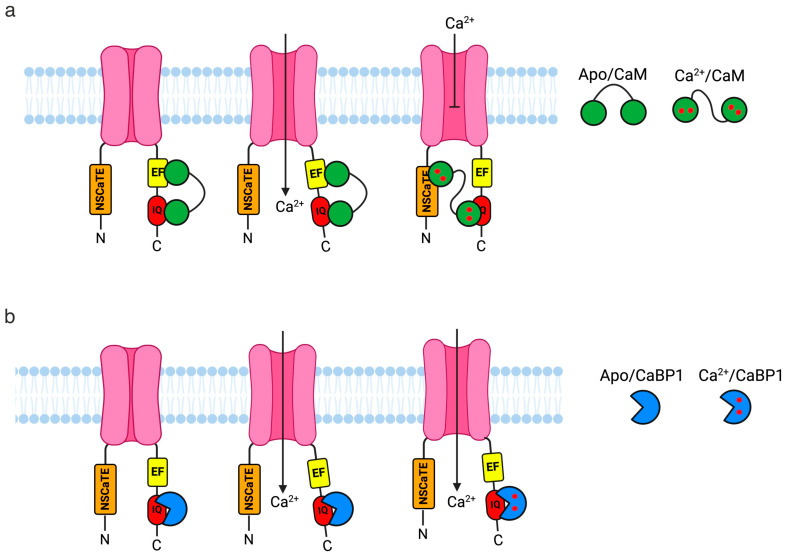
Ca^2+^-dependent inactivation (CDI) model of Ca_v_ regulation by CaM and CaBP1. (**a**) Ca^2+^-free (apo) CaM and CaBP1 compete to be pre-associated with the IQ domain (red) of the closed Ca_v_ channel (pink) under resting conditions ([Ca^2+^]_i_ = 100 nM). Next, membrane depolarisation causes an influx of Ca^2+^ through the open Ca_v_, with initial low Ca^2+^ levels meaning that the channel’s open state is stabilised by CaM. Once cytosolic Ca^2+^ reaches 1 µM, Ca^2+^ binds to CaM, which causes channel inactivation (CDI), further stabilised by Ca^2+^/CaMs N-lobe binding to the NSCaTE (orange) domain of Ca_v_1.2. (**b**) Alternatively, apo-CaBP1 is pre-associated with the IQ motif of Ca_v_ under resting conditions ([Ca^2+^] _i_ = 100 nM). Next, when cytosolic Ca^2+^ reaches higher levels due to membrane depolarisation, Ca^2+^ binds to CaBP1, which can displace CaM and reduce or abolish CDI by stabilising the open state of the channel, leading to CDF. Created in BioRender.

**Figure 4 cells-14-00152-f004:**
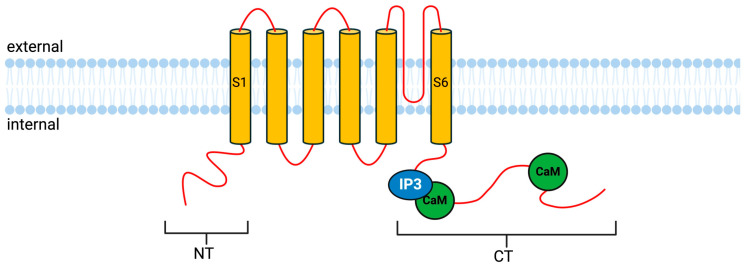
Topology of the transient receptor potential canonical receptor type 5 (TRPC5). S1–6 indicate transmembrane regions, CaM- and IP3-binding sites on the C-terminal, with the second CaM site further down. It is suggested that there are multiple sites for CaBP1 binding, with the sites being very close to or even overlapping the CaM sites, similar to Ca_v_1.2 channels. Created in BioRender.

**Figure 5 cells-14-00152-f005:**
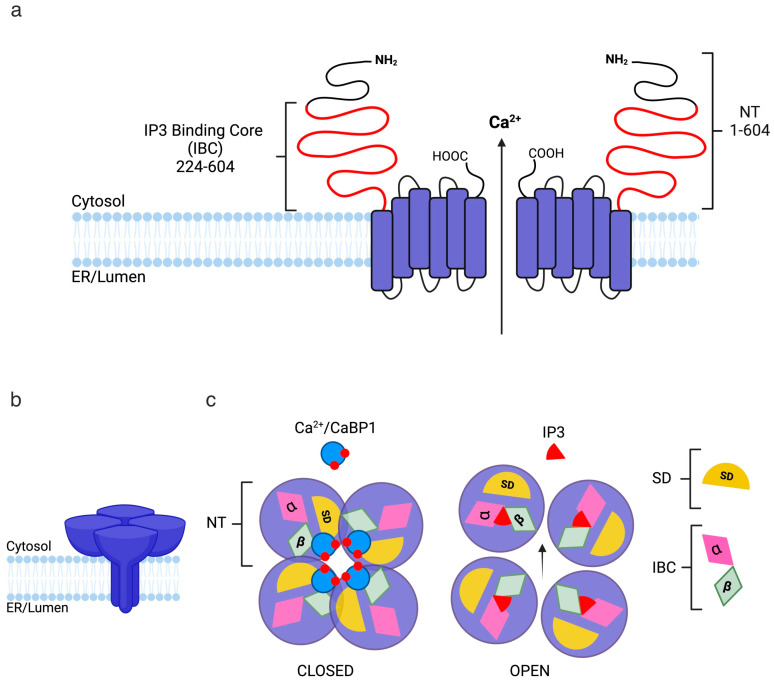
Topology of the inositol triphosphate receptor (IP3R). (**a**) Side-on view of two subunits of an active IP3R showing a membrane-inserted channel domain with cytosolic N terminals (NTs, residues 1–604), which contain the suppressor domain (SD, residues 1–223), IP3-binding core (IBC, residues 224–604), and the CaBP1-binding sites. (**b**) Tetrameric IP3R receptor in the ER membrane. (**c**) Closed and open state of an IP3R channel from a bird’s-eye view. The closed state is bound by four Ca^2+^/CaBP1s within the IBC, clamping intrasubunit movement between the β-domain of the IBC and the SD and stabilising the closed state. The open state is bound by four IP3 molecules allowing for Ca^2+^ movement through the channel. Created in BioRender.

**Figure 6 cells-14-00152-f006:**
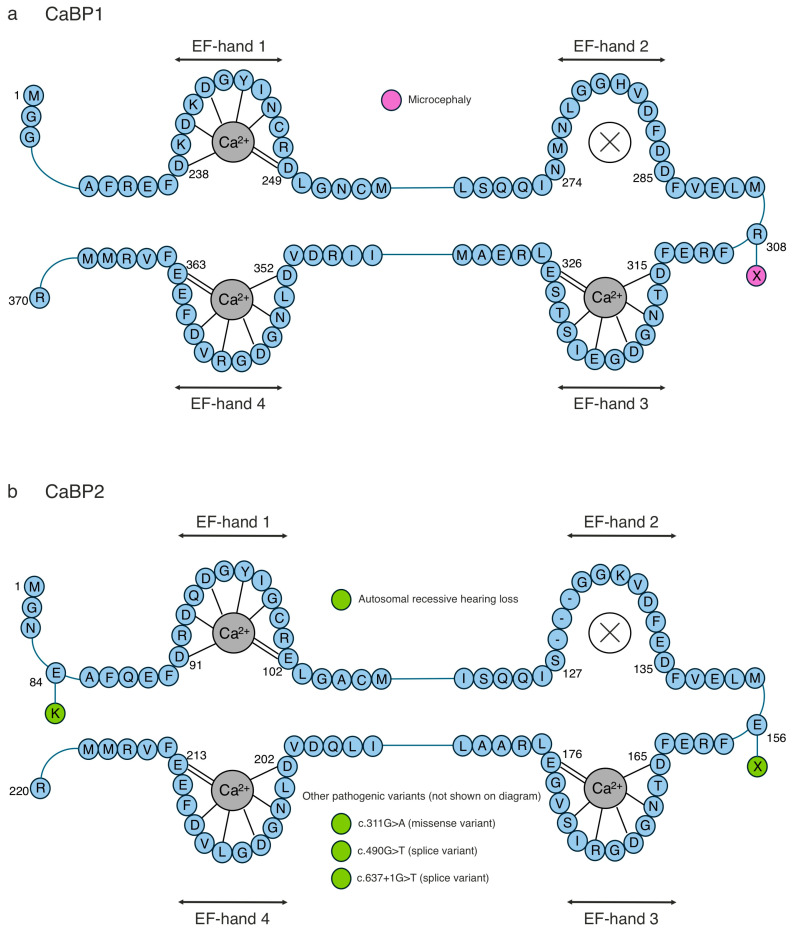
Primary structure of CaBP1–2, highlighting human pathogenic mutations. The primary structure of (**a**) CaBP1 and (**b**) CaBP2, focusing on the amino acid sequence around the four EF-hands, is shown in blue. The residues involved in Ca^2+^ coordination are indicated by dashed lines. Disease-associated mutations that have been identified in humans are illustrated as alternatively coloured circles at the relevant labelled positions. CaBP2 has been strongly linked to autosomal recessive hearing loss.

**Figure 7 cells-14-00152-f007:**
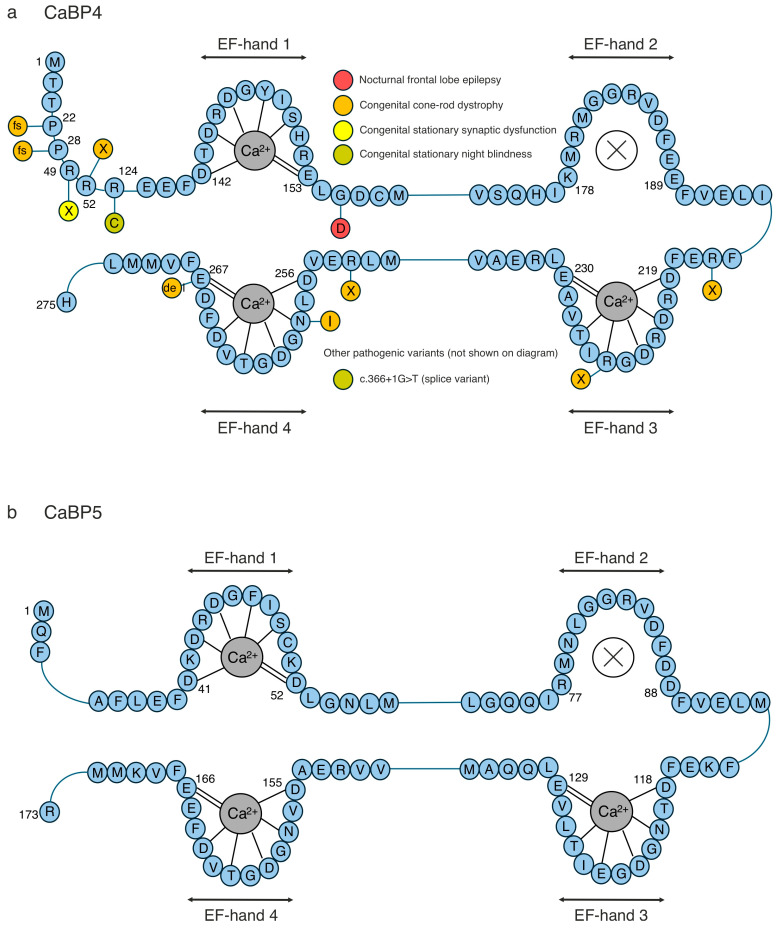
Primary structure of CaBP4–5, highlighting human pathogenic mutations. The primary structure of (**a**) CaBP4 and (**b**) CaBP5, focusing on the amino acid sequence around the four EF-hands, is shown in blue. The residues involved in Ca^2+^ coordination are indicated by dashed lines. Disease-associated mutations that have been identified in humans are illustrated as alternatively coloured circles at the relevant labelled positions. CaBP4 has the most pathogenic mutations and has been associated with several disease conditions. CaBP5 has had no relevant pathogenic mutations found.

**Figure 8 cells-14-00152-f008:**
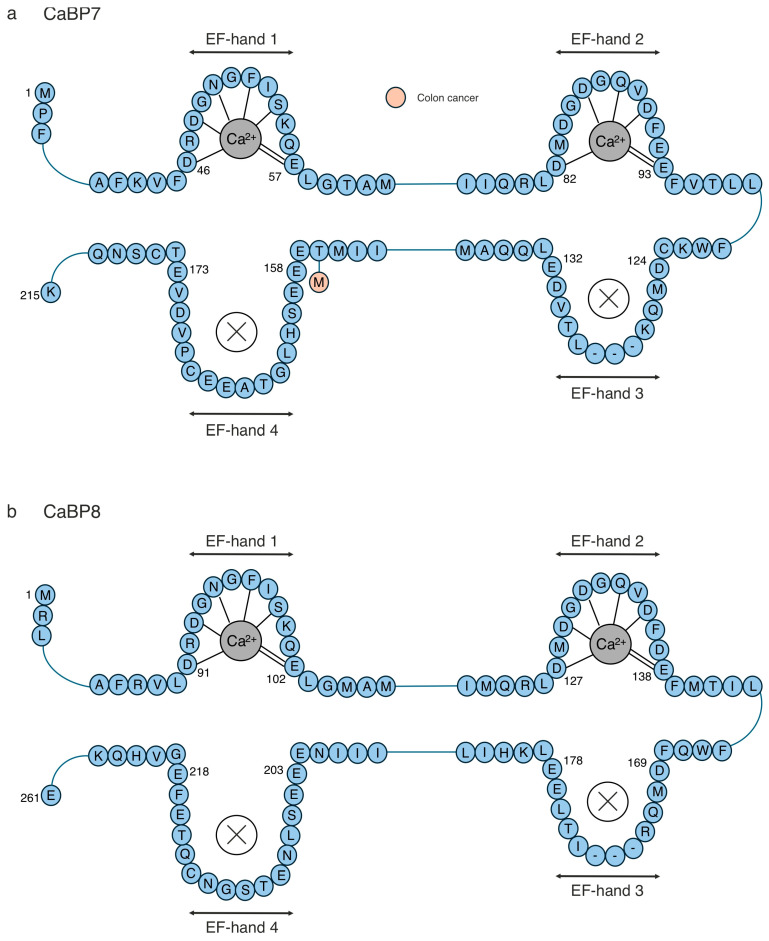
Primary structure of CaBP7–8, highlighting human pathogenic mutations. The primary structure of (**a**) CaBP7 and (**b**) CaBP8, focusing on the amino acid sequence around the four EF-hands, is shown in blue. The residues involved in Ca^2+^ coordination are indicated by dashed lines. Disease-associated mutations that have been identified in humans are illustrated as alternatively coloured circles at the relevant labelled positions. CaBP8 has had no relevant pathogenic mutations found.

**Table 1 cells-14-00152-t001:** Localisation and expression patterns of CaBPs and CaM.

Gene Name	Protein Name	Uniprot Accession Number	Subcellular Location(s)	Expression	References
*CALM1*/*CALM2*/*CALM3*	Calmodulin	P0DP23/P0DP24/P0DP25	Cytoplasm, cytoskeleton, microtubules	Cytoplasm, cytoskeleton, microtubules	[24,25,26,27]
*CABP1*	Calcium-Binding Protein 1 (Caldendrin)	Q9NZU7	Cell membrane, cytoplasm (cytoskeleton and perinuclear region), Golgi apparatus	Bipolar cells, excitatory neurons, horizontal cells	[19,24,28,29,30]
*CABP2*	Calcium-Binding Protein 2	Q9NPB3	Cell membrane, cytoplasm (perinuclear region), Golgi apparatus	Bipolar cells, horizontal cells, rod photoreceptor cells	[19,22,24,31]
*CABP4*	Calcium-Binding Protein 4	P57796	Cytoplasm, presynapse (rod spherules and cone pedicles)	Rod and cone photoreceptor cells; bipolar, horizontal, and Muller glia cells	[19,24,32]
*CABP5*	Calcium-Binding Protein 5	Q9NP86	Cytoplasm	Rod photoreceptor cells; bipolar, Muller glia, and horizontal cells	[19,24,33,34]
*CABP7*	Calcium-Binding Protein 7 (Calneuron II)	Q86V35	Golgi apparatus (trans-Golgi network membrane), cell membrane, cytoplasm (perinuclear region)	Oligodendrocyte precursor cells, rod photoreceptor cells, Muller glia cells, inhibitory and excitatory neurons. Low expression in cardiomyocytes.	[22,24,35]
*CABP8*	Calcium-Binding Protein 8 (Calneuron I)	Q9BXU9	Golgi apparatus (trans-Golgi network membrane), cell membrane, cytoplasm (perinuclear region)	Inhibitory and excitatory neurons, astrocytes, oligodendrocytes, microglial cells, oocyte germ cells	[22,24,36,37]

**Table 2 cells-14-00152-t002:** Experimentally derived structures of CaBPs and CaM in the absence of Ca^2+^.

Protein	Deposition	Species	Size of Fragments	Experimental Conditions	Comments	References
CaM	PDB 1DMO (NMR)	*Xenopus laevis*	1–148	10 mM EDTA, 10% D_2_O, 298K, pH6.0	Eight helices identified, which all adopt a regular alpha-helical conformation except for one within the N-terminal portion. Each EF-hand is composed of two helices. The first 5 N-terminal residues and last 2 C-terminal residues are not well defined. High flexibility of the linker between the lobes.	[52]
	PDB 1CMF (NMR)	*Bos taurus*	76–148	0.1 M EGTA, 10% D_2_O, 301K, pH 6.0	Secondary structures of Ca^2+^ free and Ca^2+^ bound are very similar, with the main variations seen in tertiary structure. Antiparallel arrangement of the two EF-hands. Almost all helix–helix distances are smaller in Ca^2+^ free, supporting the idea of a “closed” conformation that opens when Ca^2+^ is added.	[53]
CaBP1	PDB 2K7B PDB 2K7C (NMR)	*Homo sapiens*	16–91 96–167	5 mM MgCl_2_, 5% D_2_O, 310K, pH 7.4	First 10 residues not assigned. 95% of the backbone and >75% of side chains assigned. Closed EF-hand structures (similar to apo/CaM).	[50]
CaBP4	n.d.	*Mus musculus*	100–271	5 mM EDTA or 5 mM MgCl_2_, 10% D_2_O, 310K, pH 7.4	N-terminal region unstructured and removed. Spectra agreement between 100 and 271 and the individual lobes, suggesting that N- and C-lobes are independent domains.	[54]
CaBP7	n.d.	*Homo sapiens*	1–100	5 mM EDTA, 5 mM EGTA, 10% D_2_O, 303K, pH 6.5	Protein is folded in the absence of Ca^2+^. Large structural change compared to Ca^2+^ bound.	[55]

n.d.: not determined.

**Table 3 cells-14-00152-t003:** Experimentally derived structures of CaBPs and CaM in the presence of Ca^2+^.

Protein	Deposition	Species	Size of Fragments	Experimental Conditions	Comments	References
CaM	BMRB 51289 (NMR)	*Homo sapiens*	1–149	1 mM CaCl_2_, 8% D_2_O, 308K, pH 7.0	~96% of the backbone assigned >85% of aliphatic and aromatic side chains assigned Structure was found identical to CaM crystal structure (number of helices and binding motifs for Ca^2+^)	[57]
	PDB 1CMG (NMR)	*Bos taurus*	76–148	0.1 mM CaCl_2_, 10% D_2_O, 301K, pH 6.0	“open” conformation, with larger helix–helix distances than apo/CaM	[53]
	PDB 1CLL (X-ray)	*Homo sapiens*	1–148	50 mM MgCl_2_, 5 mM CaCl_2_, 50 mM NaOAc (pH 5.0), 17.5% (*v*/*v*) 2-methyl-2,4-pentanediol, 7.5% (*v*/*v*) ethanol	Resolution achieved for the crystal structure was 1.70 Å Dumbbell-shaped molecule with similar lobes connected by a central alpha helix Each lobe contains three alpha helices and two Ca^2+^-binding EF-hands	[59]
CaBP1	PDB 2LAN PDB 2LAP PDB 2K7D (NMR)	*Homo sapiens*	18–91 98–167 96–167	5 mM CaCl_2_, 10% D_2_O, 310K, pH 7.4	First 15 residues disordered and not assigned >95% backbone and side chains assigned Ca^2+^ bound at EF-hands 1, 3, and 4; EF-hand 1 shows closed conformation	[40,50]
	PDB 3OX5 (X-ray)	*Homo sapiens*	15–166	125 mM KCl, 0.5 mM CaCl_2_, 5 mM Tris, pH 7.4, 0.95 M (NH_4_)_2_SO_4_, 1–2% 1,2-propanediol, 0.05 M sodium citrate (pH 5.5), 20 °C	C-lobe has open conformation N-lobe in apo-form even when mM Ca^2+^ present N-lobe shows no metal ion in either EF-hand	[49]
CaBP4	PDB 2M29 PDB 2M28 (NMR)	*Mus musculus*	100–200 198–271	5 mM CaCl_2_, 10% D_2_O, 310K, pH 7.4	First 20 residues disordered Ca^2+^ bound at EF-hands 1, 3, and 4; EF1 shows closed conformation	[54]
CaBP5	PDB 4OCI (X-ray)	*Entamoeba histolytica*	1–146	2.8 M sodium acetate, 0.1 M bis-tris propane, pH 5.5, 16 °C.	Resolution for crystal structure is 1.9 Å Two globular lobes containing four alpha helices connected by loops Only one Ca^2+^ bound (N-terminal lobe)	[60]
CaBP7	PDB 2LV7 (NMR)	*Homo sapiens*	1–100	No CaCl_2_ added, 10% D_2_O, 303K, pH 6.5	NTD was Ca^2+^ bound without the use of external Ca^2+^ Ca^2+^-bound NTD has open conformation Ca^2+^-bound NTD shows larger solvent-exposed hydrophobic surface than CaM or CaBP1	[55]

**Table 4 cells-14-00152-t004:** Summary of the known interactions between ion channels and CaBPs.

	Channel
Protein	Ca_v_1.2	Ca_v_1.3	Ca_v_1.4	Ca_v_2.1	IP3R	TRPC5	PIEZO2	References
CaM	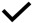	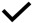	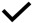	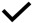	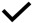	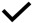		[79,85,105,122,129]
CaBP1	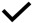	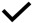		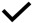	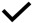	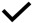	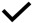	[80,81,82,84,92,115]
CaBP2		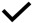						[31,94]
CaBP4	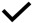	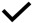	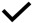					[92,96]
CaBP5	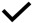							[34]

## Data Availability

Not applicable.

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
