# Peer review of "Structure–Function Diversity of Calcium-Binding Proteins (CaBPs): Key Roles in Cell Signalling and Disease"

_cells, 2025, doi:10.3390/cells14030152_

Round 1
Reviewer 1 Report
Comments and Suggestions for Authors
The Authors of this review entitled “Structure-function diversity of calcium-binding proteins 2 (CaBPs): key roles in cell signalling and disease” focus on the Ca²⁺ binding protein (CaBP) family as calcium (Ca²⁺) ions are essential cellular messengers that regulate various cellular activities in response to both intrinsic and extrinsic stimuli, such as hormonal signals, neurotransmitters, and temperature changes. Intracellular Ca²⁺ concentrations are finely regulated, as sustained increases can trigger harmful effects like organelle remodeling, reactive oxygen species production, and excitotoxicity, particularly in neurons and vascular smooth muscle cells. Cells manage Ca²⁺ levels through two main mechanisms: release from intracellular stores via channels like IP₃Rs and RyRs, and influx from the extracellular space through channels like VGCCs and LGICs. In the present review, the Authors emphasizing CaBPs roles, localization, structure, and clinically relevant mutations to improve understanding of their lesser-known members.
I believe that the topic addressed in the review deserves to be published, as little is currently known about CaBPs and the available information is often scattered throughout the scientific literature. Therefore, this work could represent a valuable and comprehensive review of what is known on this subject.
I have a general comment on the figured that are proposed:
- The text does not include any references to the figures provided at the end of the manuscript. Therefore, it is necessary to clearly indicate references to each figure within the text.
- The review could benefit from a cartoon illustrating the structure of CaBPs, as well as a visualization highlighting their structural similarity with CaM, rather than focusing on the topology of two of the channels with which the CaBPs interact.
- panel A is associated only with the Apo/CaM protein, whereas the legend suggests that it could refer to either Apo/CaM or CaBP. Additionally, the acronym NSCaTE is not defined.
- Figures 4 and 5 have low image quality, and the text is in a font that is too small, making it unreadable.
Regarding the Tables: I would suggest to look for a better impagination of the table titles (first raw)
Regarding the text:
- Is anything known about interaction of CaBP with RyRs?
- Line 253 introduce the acronym VGCCs for Voltage Gated Calcium Channels
- Lines 256-259: This part of the text seems to return to the introduction of the topic. I suggest removing it.
- Line 259: use the acronym VGCCs
- Line 262: the symbol CaV has never been introduced before, please introduce it
- Line 288 “C-lobe of interact”…something is missing, please complete
- Line 353: at line 80 it is stated that CaBP1 and caldendrin are the same protein. Here it seems that they are two different proteins. Please clarify and if they are the same protein, use CaBP1 as in the other paragraph.
- Line 373, see previuous comment
- Line 375-380, a cartoon may help
- Line 381-385. Please comment whether CaBP1 has a similar role as FKBP hai in RyR.
- Line 428: inroduce here the significance of the nonsense mutation, “The CaBP1 R308X nonsense mutation, which results in a truncated version of the protein….” So that it can be avoided in line 435
In general, carefully review the text for typos, as there are spaces where they should not be.
Reviewer 2 Report
Comments and Suggestions for Authors
In this review, “Structure-function diversity of calcium-binding proteins (CaBPs): key roles in cell signalling and disease”, the authors briefly introduced the structure-function analysis of the CaBP family, highlighting key similarities and differences both within the family and in comparison to calmodulin (CaM). Calcium ions serve as essential and highly versatile cellular messengers across all cell types, regulating diverse activities in response to both extrinsic and intrinsic stimuli. The focus of this review is on CaBPs, which may provide predictions into the characteristics and mechanisms of the lesser studied members of the CaBP family. Thus, this topic is interesting. However, the description and logic of the article should be sorted out and enhanced. Moreover, the authors did not well discuss its potential clinical application, extended diagnosis, prognosis and therapy.
Specific comments:
1. The title of this review is “Structure-function diversity of calcium-binding proteins (CaBPs): key roles in cell signalling and disease”. However, the authors did not well discuss the function role in cell signaling and disease. The authors should provide more information in cell signaling and show a related figure. The authors just showed the “6. CaBPs dysfunction is associated with neuronal disorders”, no other disease?
2. In this review, the authors did not well discuss its extended diagnosis, prognosis and therapy.
3. In this review, the aims of this review are great and profound, but the content was simply a summary. However, the logic is not well, and it is hard to get the key point. For example, the introduction of Transient receptor potential canonical type 5 (TRPC5) is not key point of this review, how can CaBPs regulate TRPC5 is important. More detailed mechanistic insights into how specific CaBPs contribute to calcium regulation would strengthen the review.
4. The authors should give the full name for every abbreviation in the first appearance.
Reviewer 3 Report
Comments and Suggestions for Authors
This study introduces the reader to a new population of less studies Ca-binding proteins CaBPs. The proteins are compared to the well know CaM (thus it seams this is not a Ca-buffering protein although this is not directly stated) and an extensive review of the knowledge of their structure is given after which the interaction with a few ion channels is shown. Finally, a summary is given for the role of known mutations in these proteins for different diseases.
Line 43: Is ref 3 adequate (plant cell!) considering the focus of paper?
Line 51: Why is ER not mentioned here? Alos in lines 57-58 just #intracellular stores” (not ER) are mentioned with receptors (IP3R, RyRs)
Line 64-64: why just CaBP5 is emphasized here?
Line 76: EF-hand motif should be explained on first mention maybe with reference to Fig 4 & 5.
Chapter 3 – It would ease reading and relate better to the title if the structural part would be already connected to some functional significance. Some general structural scheme of CaBPs would help in following this text.
Line 261: Why the “three main types of VGCCs” are not mentioned??
Line 336-337: if CAM overexpression has no effect what is then the significance of its binding?
Line 353: Caldendrin is introduced for the first time without any explanation of its nature and function.
Line 416: reduction of PI4P production should be stated above first. And what about the effect on Golgi function?
No Figures are cited in the text!?
Figure 1: There are no ABCD labels in the figure. Terminal’s binding sites should be at least mentioned.
Figure 2: Where in the scheme is the IP3 binding site?
Figure 3: There are no A & B labels in the figure.
Figure 4: Line 523-524 – Why is CaBP5 then in the Figure with this title?
Figure 5: Same for CaBP8 as for CaBP5 in Fig.4.
MINOR
Line 82: CaBP3 – if the gene is mentioned the symbol should be in italics
Table 1: Muller cells – use capital M. “also known as” is redundant
Line 313: beginning of line is repetition from line 279
Line 352: cochlea
Line 395: term calneurons was already introduced at lines 81-82 (without abbreviations which don’s seam necessary)
Line 478: “trafficking” of what?
TYPOS
There are a few unfinished verses:
Line 288: “C-lobe of…”
Line 432-3: “CaBP2 regulates auditory…”
Line 447: “There are no …”
